# A 500 kHz to 150 MHz Multi-Output Clock Generator Using Analog PLL and Open-Loop Fractional Divider with 0.13 μm CMOS

**Junting Jin** [1,2], **Yuhua Jin** [3] and **Yebing Gan** [3,*]

1. Institute of Microelectronics, Chinese Academy of Sciences, Beijing 100029, China; jinjunting20@mails.ucas.ac.cn
2. School of Microelectronics, University of Chinese Academy of Sciences, Beijing 100049, China
3. Hangzhou Zhongke Microelectronics Co., Ltd., Hangzhou 310053, China; jinyuhua@hzzkw.com
* Correspondence: ganyebing@hzzkw.com

**Abstract:** Clocks are widely used in multimedia and electronic devices, and they usually have different frequency demands. This paper presents the design of a multi-output clock generator using an analog integer-N phase-locked loop (PLL) and open-loop fractional dividers. The PLL based on a three-stage ring voltage-controlled oscillator (VCO) is used to transform the lower frequency reference into a high-frequency intermediate clock (600 MHz–900 MHz). Then, relying on the open-loop fractional divider, a wide frequency range of 500 kHz to 150 MHz can be generated. Due to the open-loop control characteristic, the clock generator has instantaneous frequency switching capability. In addition, phase-adjusting circuits added to the divider greatly improved the jitter performance of the output clock; its RMS jitter is 5.2 ps. This work was conducted with 0.13 μm CMOS technology. The open-loop divider occupies an area of 0.032 mm$^2$ and consumes 7.7 mW from a 1.2 V supply.

**Keywords:** PLL; ring VCO; fractional divider; frequency switching; bandwidth; deterministic jitter

## 1. Introduction

Modern communication, network, and video hardware devices use a wide variety of processors, FPGAs, and memory to perform the tasks and processes required by end applications [1,2]. The timing architecture in these applications is becoming increasingly complicated due to the growing level of integration required in new designs. To meet the unique requirements for each hardware design, a multi-output clock generator is proposed. By providing good frequency flexibility, it can be applied to replace fixed-frequency clock generators and discrete crystal oscillators.

Usually, the fractional-N phase-locked loop can also be used to meet diverse clock requirements due to its any-frequency output ability [3–5]. However, the PLL bandwidth limits the maximum speed of frequency switching. While a processer may require a fast frequency switching ability to support power-saving techniques, such as dynamic frequency scaling (DFS), modern multicore processors also require dynamically adjustable per-core-clock generators that need to provide fast frequency switching with no frequency overshoot and tight settling time constraints [6,7]. To solve this problem, open-loop modulation has been proposed. Several methods, such as multi-phase switching, digitally controlled delay lines (DCDL), and digital-to-time converter (DTC), have been implemented to achieve this goal. Multi-phase switching and DCDL have precise modulation depth and excellent switching speed, but they suffer from large deterministic jitter. By using a high-resolution DTC, low-jitter performance can be achieved. However, to eliminate jitter caused by quantization noise, an additional calibration cell is needed to calibrate the gain of the DTC, which increases the complexity of the circuit [8,9].

Without the bandwidth limitation, open-loop dividers have faster frequency switching, and frequency overshoot can be avoided. In view of the advantages of open-loop

modulation, a new architecture of an open-loop divider is proposed. It contains a first-order delta-sigma modulator (DSM), a multi-modulus divider (MMD), a phase error calculator, and a phase-adjusting cell. Unlike all digital methods above, the phase-adjusting cell causes a phase delay by changing the charging current of a capacitor to eliminate the phase error. By comparison, this method is simpler. It can achieve excellent jitter performance and switching speed at the same time.

The rest of this paper is organized as follows. The design details of this work are presented in Section 2. The simulation results of the circuits are shown and analyzed in Section 3. The results are illustrated and discussed in Section 4. Finally, conclusions are drawn in Section 5.

## 2. Materials and Methods

The whole architecture of the clock generator is shown in Figure 1. The output of the PLL can be delivered to several fractional dividers to provide different frequency outputs.

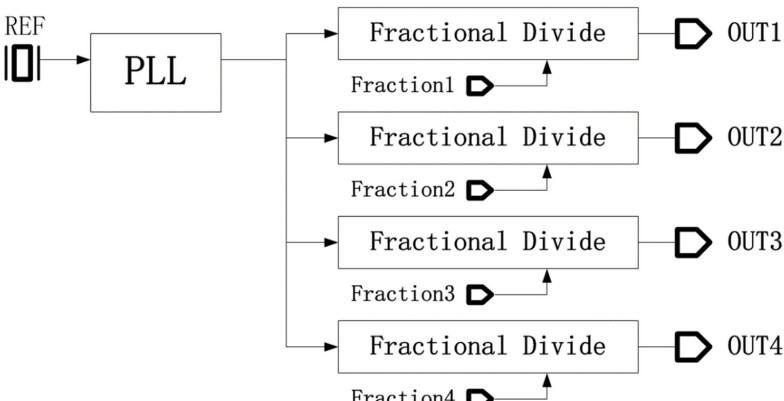

**Figure 1.** The whole architecture of the clock generator.

### 2.1. PLL Design

A block diagram of the PLL is shown in Figure 2. Since the PLL has been widely studied [10–12], here, only a brief description of the circuit design is given.

First, a phase-frequency detector (PFD) uses two D switches to compare the input clocks and sends UP and DOWN signals to control the charge pump cell. When both the UP and DOWN outputs become high, the AND gate output will simultaneously be high, resetting both switches [11]. Regarding the charge pump (CP) and the passive low-pass filter (LPF), they are applied to transform the phase differences into a tuning voltage. At the same time, high-frequency noise resulting from the blocks before the LPF will be filtered. Through the up and down currents supplied by the CP, the tuning voltage changes gradually. Until the feedback frequency and the reference frequency are equal, the tuning voltage becomes constant. At this time, the VCO will generate a stable high-frequency clock. The frequency can be given as:

$$f_{vco} = f_{ref} \times N \tag{1}$$

Regarding the cost savings, a ring VCO is applied to the PLL, whose area expense is smaller. Figure 3 shows the schematic of the three-stage ring VCO. Here, when the intrinsic gain of NM1 is big enough, the input Vc can transform into a current linearly. Then the drain current of NM1 will provide a bias current for the delay cell through the current mirror structure. The tuning curve of the VCO is chosen with the capacitor arrays. The automatic frequency control (AFC) is achieved using Verilog code.

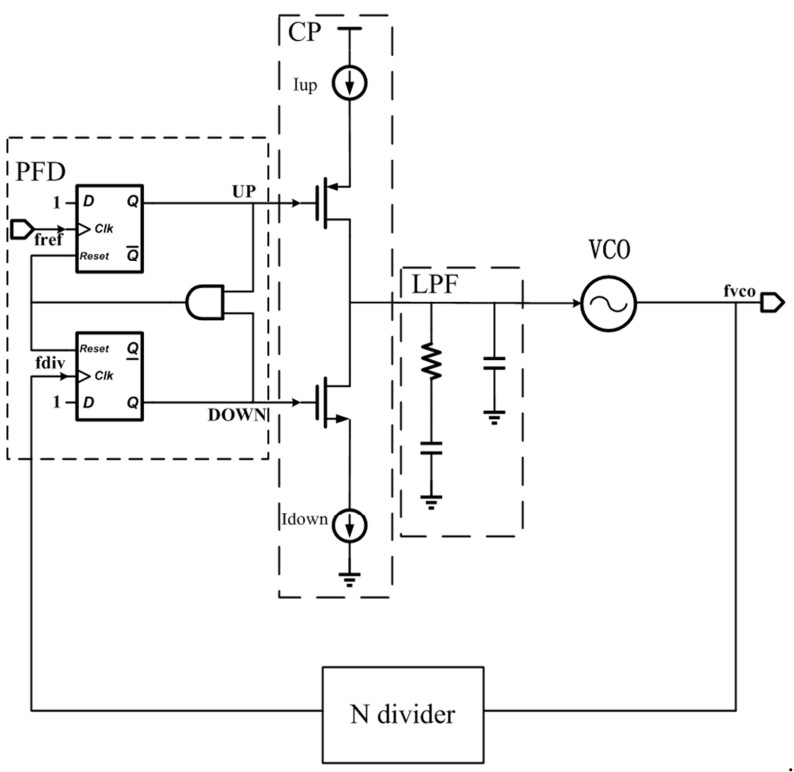

**Figure 2.** Block diagram of PLL.

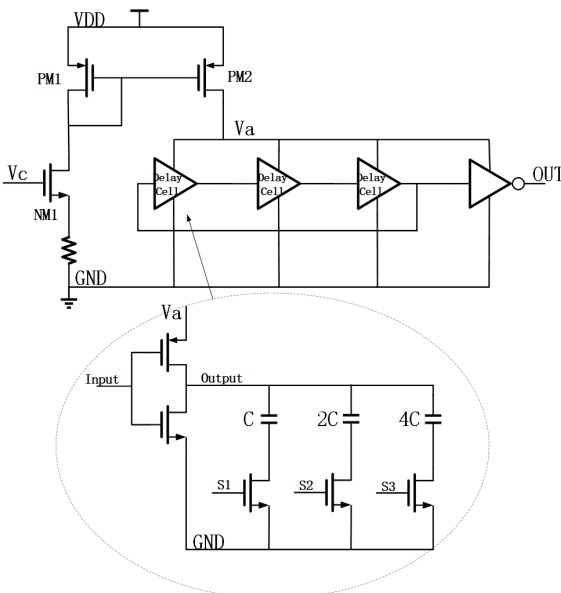

**Figure 3.** Schematic of VCO.

### 2.2. Open-Loop Fractional Divider Design

A block diagram of the open-loop divider we used is shown in Figure 4. To eliminate quantization errors caused by the DSM, the relative phase difference in the MMD is calculated by comparing the fraction and integer generated by the DSM, and a control signal is transferred to the phase-adjusting cell. Then the phase of the desired output clock will be dynamically adjusted to match the ideal clock waveform. Based on this architecture, each output clock can be selected to generate any frequency from 500 kHz to 150 MHz.

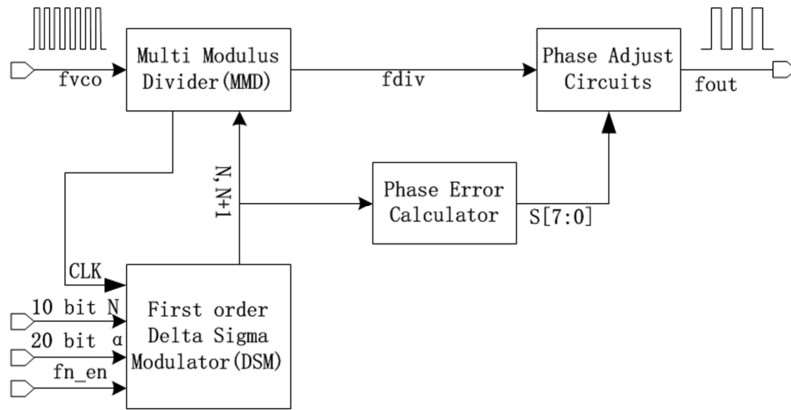

**Figure 4.** Block diagram of open-loop divider.

2.2.1. Multi-Modulus Divider

The schematic of the MMD is shown in Figure 5. It is based on a divide-by-2/3 cell and a *P* counter [13]. The *P* [9:0] is obtained from the DSM block. The output frequency of the VCO (fvco) is first sent to a divide-by-2/3 cell. If the *P* [0] is 0, then the divide-by-3 function fails. Otherwise, when the *P* counter decreases to zero, it generates a high-level signal Ld, which resets the *P* counter and actives the divide-by-3 function. At the same time, the *P* counter sends an output divide signal to the divide-by-2 cell. It is used to confirm the 50% duty cycle. Thus, the division ratio is always an even integer. Finally, the output frequency division ratio can be described as:

$$DR = 2 \times [2 \times (P[9:1] + 1) + P[0])] = 4P[9:1] + 2P[0] + 4 \tag{2}$$

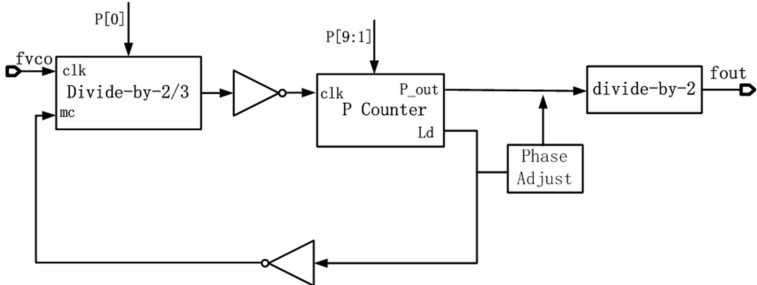

**Figure 5.** Schematic of MMD.

2.2.2. Phase-Adjusting Circuits

Figure 6 shows how to achieve a division ratio of 4.25 using a 4/5 dual-mode divider. It can be realized by dividing the input by 4 for three cycles and by 5 for one cycle in a repetitive manner [9]. We found that the deterministic jitter was nearly as large as the $T_{IN}$. For better application, it must be kept as small as possible. As a result, phase-adjusting circuits were added to improve it.

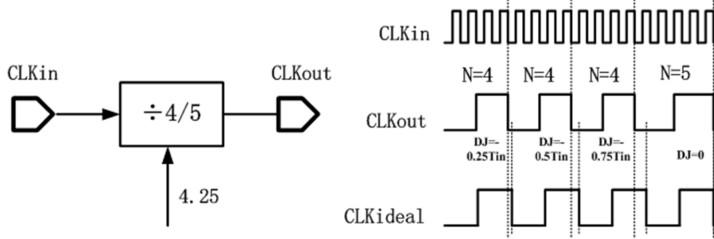

**Figure 6.** Block and timing diagrams of a fractional divider.

As shown in Figure 5, phase-adjusting circuits change the time when the rising edge of the P counter divide signal arrives. In this way, the phase of the final output clock can be matched with the ideal clock.

The specific implementation can be seen in Figure 7. As for the tail current sources (NM1 and NM2), there are eight groups whose transistor multiplier numbers decrease by multiples of 2. An RC (R1, C1) filter was added for reducing the bias current noise and stabilizing the gate voltage. In this manner, the current sources can be more accurate. The switching of PM1 and NM1 is controlled by digital circuits. We can see that if PM1 is on and NM1 is off, then the Va voltage will be increased to VDD. After that, if NM1 is on, the Va voltage will decrease to the previous value. The time when the Va changes influences the output of the P counter through a logic transform circuit.

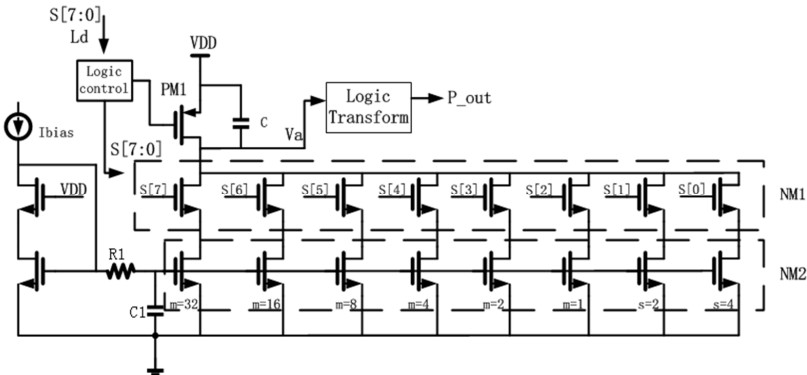

**Figure 7.** Schematic of phase-adjusting circuit.

However, the discharging time of the capacitor is related to the current given by the tail current sources. At this time, the current through the NMOS device NM1 is considered $I = C \times dU/dt$, where $I$ represents the total currents when all the sources are on. At this time, the slope of the voltage decreasing can be shown as follows:

$$k0 = \frac{dU}{dt} = \frac{I}{C} \tag{3}$$

If some of the current sources are turned off, the decreasing Va speed will be slower. The slope can be described as:

$$k1 = \left\{ s\,[7:0] \cdot \left( \frac{I}{2}, \frac{I}{2^2}, \frac{I}{2^3}, \frac{I}{2^4}, \frac{I}{2^5}, \frac{I}{2^6}, \frac{I}{2^7}, \frac{I}{2^8} \right)^T \right\} / C \tag{4}$$

where $s\,[7:0]$ is obtained from the phase error calculator cell. The rising edge mentioned above will be delayed accordingly. It can be calculated as:

$$T_{delay} = t1 - t0 = \frac{\Delta U}{k1} - \frac{\Delta U}{k0} \tag{5}$$

where $\Delta U$ is the variation of the capacitor voltage. What is more, to better cancel out deterministic jitter, the time that the current sources are turned off is set as $T_{in}$. Under this ideal circumstance, the delay time can be expressed as:

$$T_{delay} = t1 - t0 = \left( T_{in} + \frac{\Delta U - k1 * T_{in}}{k0} \right) - \Delta U/k0 = \left( 1 - \frac{k1}{k0} \right) T_{in} \tag{6}$$

In calculating the delay times when only one current is turned off, they can be listed as $T_{in}/2$, $T_{in}/2^2$, $T_{in}/2^3$, $T_{in}/2^4$, $T_{in}/2^5$, $T_{in}/2^6$, $T_{in}/2^7$, and $T_{in}/2^8$ by size. As a result, the

best phase resolution can be regarded as $T_{in}/2^8$. This is partly determined by the input frequency. Thus, it is better to choose a higher frequency input.

NM1 is not an ideal switch. It causes errors for the compensation of deterministic jitter.

### 2.2.3. DSM and Phase Error Calculator

This calculation is accomplished using Verilog code design as well. According to the designed phase-adjusting circuits above, a first-order DSM was chosen so that deterministic jitter will not exceed $T_{in}$ [14]. The input division ratio signal is composed of an 11-bit integer and a 20-bit fraction. The integer division ratio (N) ranges from 6 to 1800 with the addition of the fractional part α, resulting in a division ratio of N + α. Additionally, an enabled control signal fn_en is included. It is a signal that controls whether the fractional division function is enabled. When the signal is set to zero, the fractional division function will turn on. Finally, the input clock signal is generated from the MMD block, which makes the division ratio change synchronously.

The output of the DSM is a dynamic number that switches between two consecutive integer values. It is sent to the MMD and the phase error calculator cell. As can be seen, since the division ratio only switches between N and N + 1, deterministic jitter can be easily calculated as $\alpha T_{IN}$ and $(1-\alpha)T_{IN}$. To cancel out the jitter, it was necessary to make the phase-adjusting circuits generate opposite delay times accordingly. For example, if the divide numbers change to N, N, and N + 1, the corresponding delay times would be $\alpha T_{IN}$, $\alpha T_{IN}$, and $-(1-\alpha)T_{IN}$. To accomplish this, the calculator controls the switches of the current sources in the phase error adjusting circuits dynamically. Here, a simple algorithm is applied.

## 3. Results

### 3.1. Simulation Results

#### 3.1.1. Tuning Range of VCO

The tuning range and the phase noise analysis of the three-stage ring VCO are shown in Figure 8. The target frequency of 600–900 MHz is achieved by controlling the switch capacitor arrays. The combined values of the capacitor and controlled voltage decide the value of output frequency.

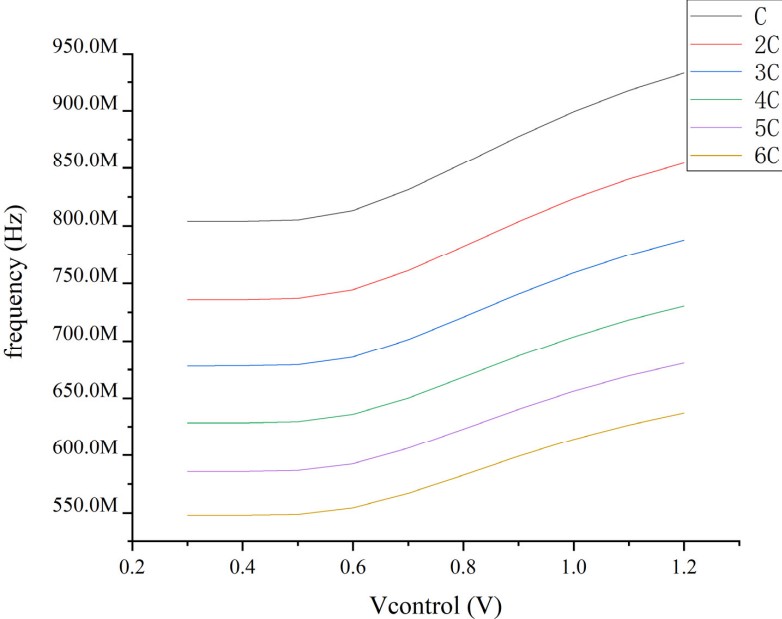

**Figure 8.** The tuning range of VCO.

### 3.1.2. Whole Performance

Figure 9 shows the output results of the clock generator. For the PLL part, we provided an input frequency of 25 MHz and a division ratio of 32. Then, using the AFC algorithm and lock process of the PLL, its output frequency is 800 MHz. For the open-loop divider, we set the fractional ratio as 30.375. Then, the $P$ [9:0], which is the output of the DSM, switches between 0000001101 and 0000001110, the according division ratio switches between 30 and 32, and the control signal S [7:0] and the output of the $P$ counter change synchronously. Similar to Figure 6, when the output of the DSM is 0000001101, a deterministic jitter of $-0.375T_{in}$ appears, and an equivalent $T_{delay}$ will be generated to offset it. In the same way, a deterministic jitter of $0.625T_{in}$ will be offset when the output of DSM is 0000001110. An output frequency of 26.337 MHz can finally be obtained.

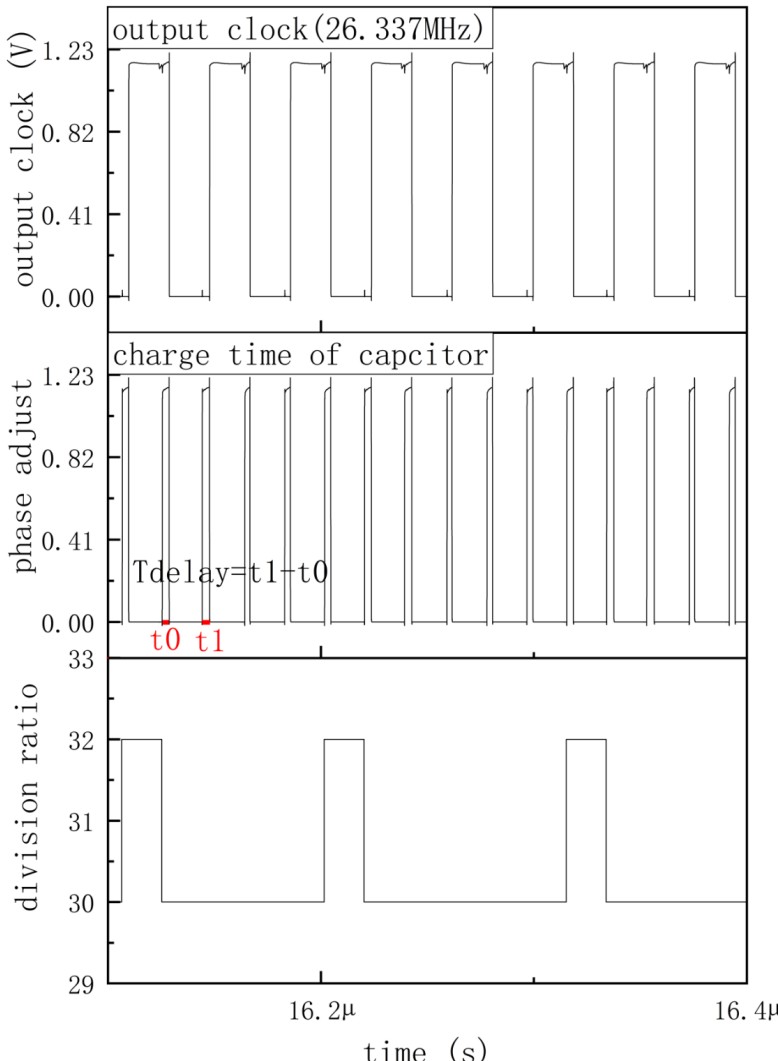

**Figure 9.** Output results of the clock generator.

### 3.1.3. Frequency Switching Speed of Open-Loop Fractional Divider

To test the fast-switching ability, we changed the N + $\alpha$ value from 30.375 to 190.875. The results can be seen in Figure 10. The switching time was less than 50 ns within a frequency step of about 28 MHz. Once a fixed input frequency is ready, the divider can generate output and switch instantly.

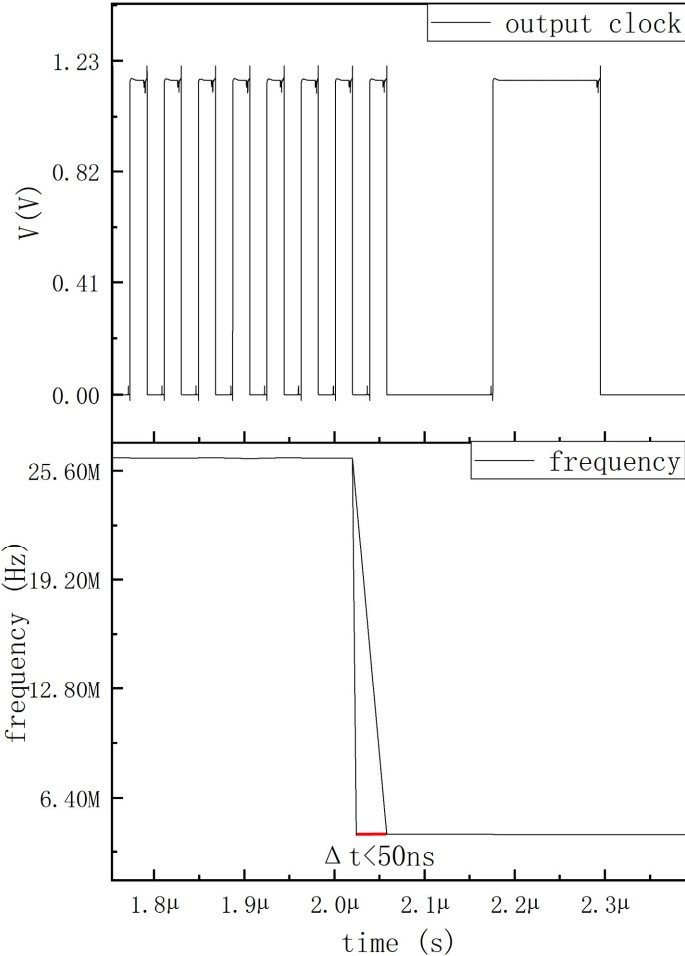

**Figure 10.** Simulation results of frequency switching.

### 3.2. Test Results

### 3.2.1. Output Clock

A 27 MHz crystal oscillator was used for the reference input clock of the analog PLL for the test. The integer division ratio was set as 24 so the output frequency of the PLL was 648 MHz. The division ratio of the open-loop fractional divider was 15.75. Finally, a clock with a frequency of 41.14 MHz was obtained. We used an oscilloscope to observe its waveform, which is presented in Figure 11. The brand of the oscilloscope was TEXTRONIX and the product model was TBS2000B.

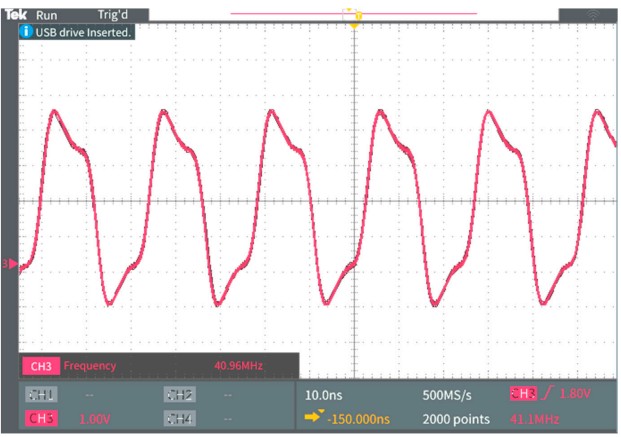

**Figure 11.** Diagram of the output clock.

### 3.2.2. RMS Jitter

As mentioned above, we obtained a further measurement of the RMS jitter with a spectrum analyzer. The test result is presented in Figure 12. Test results of RMS jitter. We can see that the carrier frequency of the output clock was about 41.14 MHz and its RMS jitter was 5.189 ps. Moreover, phase noise at the 1 MHz offset was about −133 dBc/Hz. The brand of the spectrum analyzer was Agilent and the product model was N9020A. The measurement setup can be seen in Appendix A.

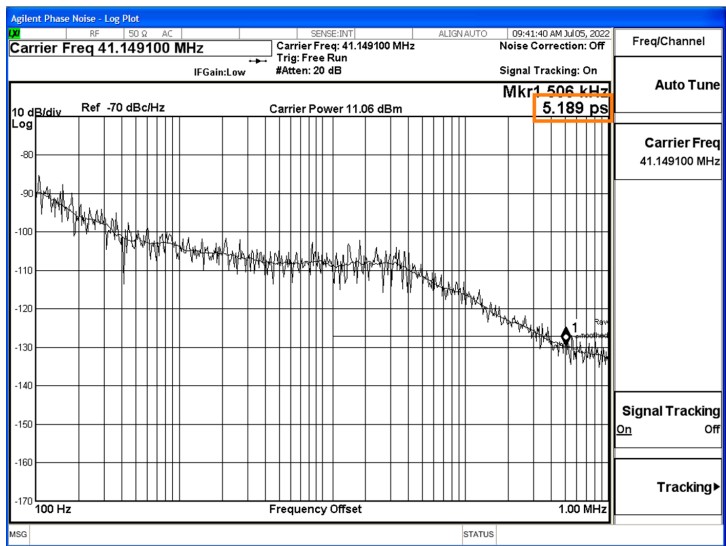

**Figure 12.** Test results of RMS jitter.

## 4. Discussion

Figure 13 shows a die photograph, which displays the PLL and the open fractional divider. We can see that the area of the open fractional divider is about 0.032 mm$^2$ (245 μm × 130 μm).

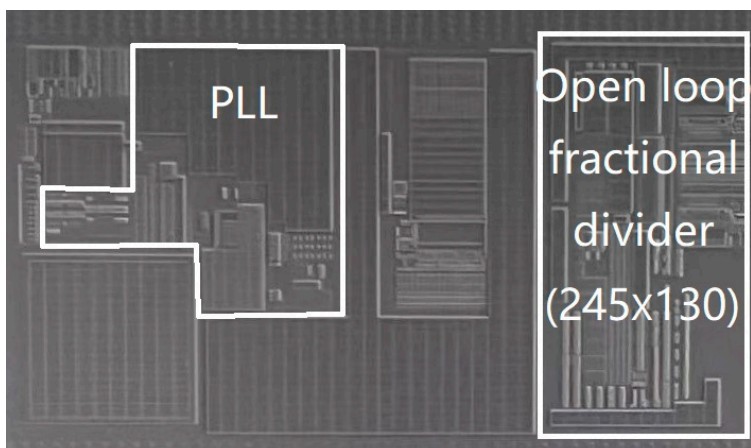

**Figure 13.** Die photograph.

The PLL design is based on a three-stage ring VCO. It uses a 27 MHz reference frequency and provides a 600–900 MHz output. Then, using an open-loop fractional divider, a clock with a frequency range of 500 kHz–150 MHz can be obtained. Through phase-adjusting circuits, the deterministic jitter of DSM is greatly suppressed. Table 1 shows a comparison of this architecture with others.

**Table 1.** Summary and comparison with different articles.

| | [9] | [15] | [16] | [17] | This Work [1] |
|---|---|---|---|---|---|
| Architecture | Fractional-N divider (DTC) | Fractional-N PLL | Fractional-N divider (DCDL) | Delay-locked loop (DCDL) | Fractional-N divider (analog phase-adjusting cell) |
| Technology (nm) | 65 | 65 | 65 | 180 | 130 |
| Supply (V) | 0.9 | 0.65 | 1.2 | 1.8 | 1.2 |
| Input frequency (MHz) | 5000 | 60 | 1270 | NA | 600–900 |
| Output frequency (MHz) | 20–1000 | 1000–1500 | 180–1270 | 60–1100 | 0.5–150 |
| RMS jitter (ps) | 1.44 | 14.0 | 12.8 | 1.4 | 5.2 |
| Instantaneous switching | Yes | No | Yes | No | Yes |
| Power consumption | 3.2 mW | 1.85 mW | 19.8 mW | 23 mW | 7.7 mW |
| Area (mm$^2$) | 0.017 | 0.23 | 0.044 | 0.066 | 0.032 |

[1] Only open-loop divider was considered when comparing the supply, power consumption, and area.

It can be seen that all designs can achieve any frequency output in a wide range. However, the open-loop fractional divider architecture has an instantaneous switching ability. In comparison, the architecture of Ref. [9] has excellent performance in general, and an external clock source with a 200 fs RMS jitter was used in the test. However, a digital calibration unit was added to suppress the influence of the process, voltage, and temperature variations, which improved the complexity of circuit design. The architectures of Ref. [17] and Ref. [18] use inverters and switch capacitor arrays as delay cells and consume a large current. The measured RMS jitter of our work was about 5.2 ps and the power consumption was 7.7 mW. However, the phase noise performance of the PLL was poor due to structural restrictions. To some extent, the forward input clock performance limits the best performance that the open-loop fractional divider can achieve. Furthermore, owing to the open-loop architecture, the modulation bandwidth is unlimited. It is perfectly suitable for a spread spectrum clock (SSC) application. In addition, an excellent electromagnetic interference (EMI) can be achieved [18]. Thus, we will change the ring oscillators to LC oscillators for better jitter performance and add an SSC function in the future.

**5. Conclusions**

In conclusion, a multi-output clock generator is proposed to meet diverse clock requirements in SoCs. By using a group of current sources for charging the capacitor, the quantization error caused by the DSM was suppressed successfully. This new architecture of an open-loop divider achieves excellent jitter performance and instantaneous frequency switching.

**Author Contributions:** Conceptualization, J.J., Y.J. and Y.G.; methodology, J.J., Y.J., and Y.G.; software, J.J.; validation, J.J. and Y.G.; formal analysis, J.J.; investigation, J.J.; resources, J.J.; data curation, J.J. and Y.G.; writing—original draft preparation, J.J.; writing—review and editing, J.J.; visualization, J.J.; supervision, Y.G. and Y.J.; project administration, J.J., Y.J., and Y.G.; funding acquisition, Y.G. All authors have read and agreed to the published version of the manuscript.

**Funding:** This research received no external funding.

**Institutional Review Board Statement:** Not applicable.

**Informed Consent Statement:** Not applicable.

**Data Availability Statement:** Not applicable.

**Conflicts of Interest:** The authors declare no conflict of interest.

**Appendix A**

Figures A1 and A2 show the measurement setup of the clock generator during testing.

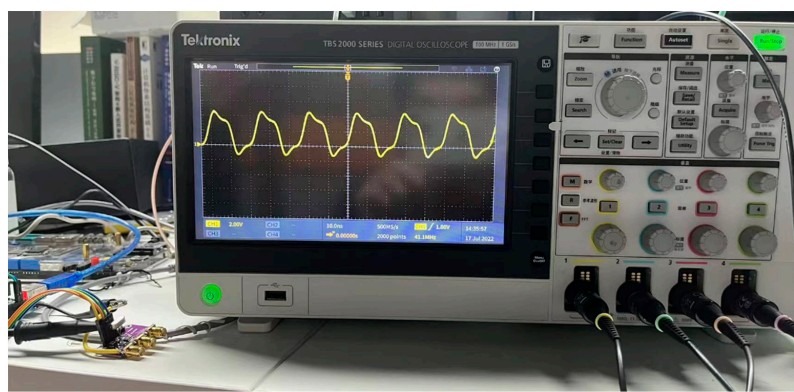

**Figure A1.** Measurement of the output clock using an oscilloscope.

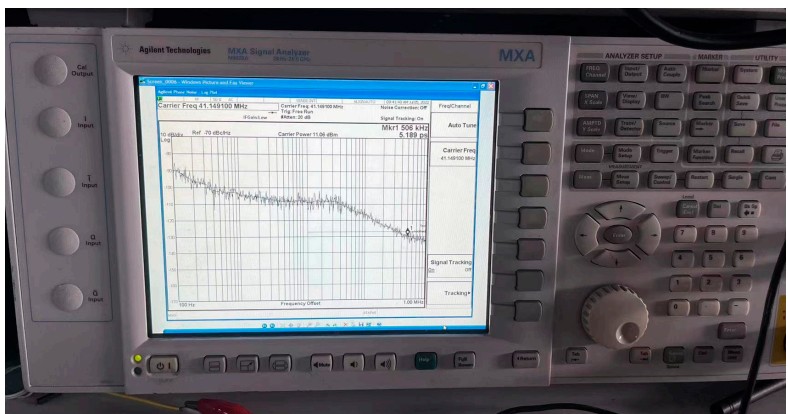

**Figure A2.** Measurement of RMS jitter using a spectrum analyzer.

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
