# Peer review of "A 500 kHz to 150 MHz Multi-Output Clock Generator Using Analog PLL and Open-Loop Fractional Divider with 0.13 μm CMOS"

_electronics, doi:10.3390/electronics11152347_

Round 1

Reviewer 1 Report

This paper discussed a clock generator that used an integer-N PLL and open-loop fractional divisions to generate a multi-output clock. Using a three-stage ring VCO, the PLL converts a low-frequency reference into a high-frequency intermediate clock (600 MHz-900 MHz). 500 kHz to 150MHz may be achieved using an open-loop fractional divider. Open-loop clock generator control permits instant frequency modification. The divider's phase adjust circuitry may decrease clock jitter to 5.2-ps jitter. The proposed clock generator is fabricated using 0.13m CMOS.  The proposed strategy for the design of the clock generator is novel. However, here are my comments:

1. In the Introduction section, the authors described their motivation for conducting the study. However, the gaps in the previous or prior works being compared in Table 1 were not introduced right away. The authors should somehow describe the strengths and weaknesses of refs. [13]-[15] and make a critical analysis of their review.

2. In line 42, the authors calculated the relative phase difference between the MMD-generated clock and what (?). It is not clarified what difference the authors are pertaining to.

3. The authors should proofread the paper first with the help of an English native speaker. There are many grammatical errors. Check for the congruence of the sentences. For instance, line 52 should be written as "The whole architecture of clock generator is shown in Figure 2... Also in line 83, the output frequency of VCO... is it fout or fvco? It should be rewritten: the output frequency (fout) of VCO or otherwise. In line 95, there is a phrase or fragment that should be avoided like: Considering about achieving a division ratio of 4.25... In line 184, the picture is drawn in Figure 11 but it is not. It should be stated as the picture is displayed or presented. Moreover, when writing numbers with corresponding units, they should be separated, e.g. 41.14 MHz instead of 41.14MHz.

4.  In line 107, the R and C in the RC filter being explained should be labeled in Figure 7 e.g. R1, C1.. A RC filter (R1, C1) is added...

5. In line 117, C*dU/dt; what is U about? I cannot find U in the schematic or its representation or description in the statements or sentences?

6. All equations are advised to be rewritten in a correct mathematical manner. Latex is strongly advised.

7. In line 166 and Figure 9, is Tdelay units in volts? Why did the authors name it Tdelay instead of Vdelay? Normally, Tdelay means delay period or delay time.

8. In line 183, what type or brand of oscilloscope did the authors use in their measurement setup? The authors should present a Figure showing the measurement setup of their fabricated clock generator chip.

9. Since their proposed clock generator was fabricated and tested, the authors should its die photo, layout, and floorplan.

10. In Figures 9 and 10, please indicate the peak voltage of the square waveforms. For instance, is the output clock peak voltage 1.22 V the same as Tdelay?

11. Avoid using black background on Figure 11. Is it possible to invert the image to an image with a white background?

12. In line 189, what type or brand of oscilloscope did the authors use in their measurement setup?

13. In Figure 12, please indicate the rms jitter by drawing a box or highlighting it.

14. The authors presented a nice comparison of their proposed work with the prior recent works on clock generators. However, they should give more details. Please write more sentences and statements describing not only the weaknesses of their proposed work over others but also the strengths of their proposed work among others.

Author Response

Point 1:In the Introduction section, the authors described their motivation for conducting the study. However, the gaps in the previous or prior works being compared in Table 1 were not introduced right away. The authors should somehow describe the strengths and weaknesses of refs. [13]-[15] and make a critical analysis of their review.

Response 1: It is my fault to skip over the difference analysis between my work and refs and I will add the description and review of refs which are compared in the discussion section.

Point 2: In line 42, the authors calculated the relative phase difference between the MMD-generated clock and what (?). It is not clarified what difference the authors are pertaining to.

Response 2 : Since the MMD-generated clock is not ideal and will cause high jitter. It should be calculating the relative phase difference between the MMD-generated clock and ideal clock and then to eliminate it. It is my unclear explanation and I will correct.

Point 3: The authors should proofread the paper first with the help of an English native speaker. There are many grammatical errors. Check for the congruence of the sentences. For instance, line 52 should be written as "The whole architecture of clock generator is shown in Figure 2... Also in line 83, the output frequency of VCO... is it fout or fvco? It should be rewritten: the output frequency (fout) of VCO or otherwise. In line 95, there is a phrase or fragment that should be avoided like: Considering about achieving a division ratio of 4.25... In line 184, the picture is drawn in Figure 11 but it is not. It should be stated as the picture is displayed or presented. Moreover, when writing numbers with corresponding units, they should be separated, e.g. 41.14 MHz instead of 41.14MHz.

Response 3: I will correct the errors and proofread the paper. However, in line 83, the output frequency of VCO is considered as fvco. I will add the explanation.

Point 4: In line 107, the R and C in the RC filter being explained should be labeled in Figure 7 e.g. R1, C1.. A RC filter (R1, C1) is added...

Response 4: I will add it.

Point 5: In line 117, C*dU/dt; what is U about? I cannot find U in the schematic or its representation or description in the statements or sentences?

Response 5: Here this formula is just a general expression to current (I). And in the following formula 3, dU/dt is same as k0. I will describe it as k0=dU/dt=I/C so that dU can be corresponding to ΔU and dt can be corresponding to t0,t1 in formula 5.

Point 6: All equations are advised to be rewritten in a correct mathematical manner. Latex is strongly advised.

Response 6: I will try that.

Point 7: In line 166 and Figure 9, is Tdelay units in volts? Why did the authors name it Tdelay instead of Vdelay? Normally, Tdelay means delay period or delay time.

Response 7: Here is my error expression. Tdelay means the difference of high level time between two periods. I will improve Figure 9 according to formula 5.

Point 8: In line 183, what type or brand of oscilloscope did the authors use in their measurement setup? The authors should present a Figure showing the measurement setup of their fabricated clock generator chip.

Response 8: I will consider to put that in appendix.

Point 9: Since their proposed clock generator was fabricated and tested, the authors should its die photo, layout, and floorplan.

Response 9: I will consider to put that in appendix.

Point 10: In Figures 9 and 10, please indicate the peak voltage of the square waveforms. For instance, is the output clock peak voltage 1.22 V the same as Tdelay?

Response 10: I will indicate the peak voltage of the output clock. However, as for Tdelay, the peak voltage is not so important and the interval between two periods is main factor. And according to point 7, I will rename it.

Point 11: Avoid using black background on Figure 11. Is it possible to invert the image to an image with a white background?

Response 11: I will try that.

Point 12: In line 189, what type or brand of oscilloscope did the authors use in their measurement setup?

Response 12: I will put a photo in appendix.

Point 13: In Figure 12, please indicate the rms jitter by drawing a box or highlighting it.

Response 13: I will do that.

Point 14: The authors presented a nice comparison of their proposed work with the prior recent works on clock generators. However, they should give more details. Please write more sentences and statements describing not only the weaknesses of their proposed work over others but also the strengths of their proposed work among others.

Response 14: I will add more details in discussion part.

Reviewer 2 Report

The authors present multi output clock generator using PLL and fractional divider. Few suggestions have been given to improve the quality of the manuscripts.

1. Introduction does not provide sufficient background study. Ref 13, 14 and 15 used in Table I have not been described accordingly in the background study. The research gap has not been defined accordingly as well. What is the difference between this work compared to previous research work? This is important to show the novelty of the work.

2. The references for the comparison in Table 1 range from 2010 -2014 which are not up to date. Suggest including more recent references.

3. This work proposes the open-loop divider to achieve faster-switching speed. Has this method never been proposed before? The switching speed  performance comparison with previous work is also missing in Table 1.

4. The novelty of the work is not clear. What is the newly proposed method? 

5. The conclusion could be enhanced by highlighting the proposed method and the achieved performance in this work.

Author Response

Point 1: Introduction does not provide sufficient background study. Ref 13, 14 and 15 used in Table I have not been described accordingly in the background study. The research gap has not been defined accordingly as well. What is the difference between this work compared to previous research work? This is important to show the novelty of the work.

Response 1: All can achieve the function of fraction divide. Ref 14 is a fractional-N PLL and its switching speed is limited by bandwith. Ref 13 and 15 use open loop divider and they can achieve instantaneous switching. However they are implemented in different ways. Ref 13 uses DTC and digital calibration units. Ref 15 uses digitally controlled delay line. They are realized in all digital aspect . However, my work calculate the phase error in digital aspect and then transformed the error into current and finally through the charge of capacitor transform the current into delay time which can adjust the phase of output clock. This part is implemented with analog circuit.

The introduction of difference may be not clarified specifically in this article and I will improve it.

Point 2: The references for the comparison in Table 1 range from 2010 -2014 which are not up to date. Suggest including more recent references.

Response 2: I will try to find more recent references.

Point 3: This work proposes the open-loop divider to achieve faster-switching speed. Has this method never been proposed before? The switching speed  performance comparison with previous work is also missing in Table 1.

Response 3: The open-loop divider has been proposed before. However, my work proposes a new implementation way which is not used before. The faster-switching speed is a characteristic of open-loop divider when compared with PLL. It may be not so important when both open-loop dividers compare. However, I will try to add the specific switching time.

Point 4: The novelty of the work is not clear. What is the newly proposed method? 

Response 4: Just as response 1 said, this article introduces a new way to realize fractional open-loop function. Unlike DTC and DCDL, they adjust phase error through all digital circuits. We calculate phase error by comparing the difference between ideal fraction and integer produced by DSM. Then transform the error to a signal which controls current and causes different charge time of capacitor. The time will eliminate the phase error finally.

Point 5: The conclusion could be enhanced by highlighting the proposed method and the achieved performance in this work.

Response 5: I will add more details about my method in conclusion section.

Reviewer 3 Report

The authors present a multi-output clock generator using analog PLL and open loop fractional divider. The manuscript is neither well organized nor well written. It must be improved under several aspects that here I try to summarize.

- It is not evident the novelty of this work. Open loop fractional divider is a known solution together with circuits for the phase noise calibrations. The authors should put the emphasis on the benefits of their technique compared to state-of-the art ones.

- The realization of a silicon prototype is mentioned only in the abstract and in the ciomparison table. No chip microphotograph is present. More details about the chip, as well as the measurement setup, are mandatory.

- The few graphs in the Results Section are insufficiently commented and do not prove completely the strengths of this work.

- I suggest to revise the structure of the manuscript (and especially Section 2). For example, it is not helpful for the reader to find first the Figure of the open loop fractional divider and then the figure of the whole architecture.

- The introduction is not enough explanatory and does not provide a sufficient description of the background. I also suggest to improve the number of references, looking also to more recent publications (the most recent reference in the manuscript is 4 years old).

- Some acronyms are not defined (e.g. PFD). Please doublechek the whole manuscript.

Author Response

Point 1: It is not evident the novelty of this work. Open loop fractional divider is a known solution together with circuits for the phase noise calibrations. The authors should put the emphasis on the benefits of their technique compared to state-of-the art ones.

Response 1: I will add more details about the difference between my work and state-of-arts and explain the benefits.

Point 2: The realization of a silicon prototype is mentioned only in the abstract and in the ciomparison table. No chip microphotograph is present. More details about the chip, as well as the measurement setup, are mandatory.

Response 2: I will add these photo in appendix.

Point 3: The few graphs in the Results Section are insufficiently commented and do not prove completely the strengths of this work.

Response 3: I will write more about the result analysis in Result Section.

Point 4: I suggest to revise the structure of the manuscript (and especially Section 2). For example, it is not helpful for the reader to find first the Figure of the open loop fractional divider and then the figure of the whole architecture.

Response 4: I consider the core point of this article is open loop divider so I put this figure at the beginning. However, it may cause inconvenience for readers to understand the Section 2, I will put this figure properly and try to adjust the structure of Section 2.

Point 5: The introduction is not enough explanatory and does not provide a sufficient description of the background. I also suggest to improve the number of references, looking also to more recent publications (the most recent reference in the manuscript is 4 years old).

Response 5: I will try to improve the number of references and find more recent publications.

Point 6: Some acronyms are not defined (e.g. PFD). Please doublechek the whole manuscript.

Response 6: It is my fault to ignore that and I will check the whole manuscript carefully.

Round 2

Reviewer 1 Report

The authors already complied with my comments. They have already revised their paper based on my comments and suggestions. The paper needs to be rechecked for English language grammar and congruity. 

Author Response

Point1: The authors already complied with my comments. They have already revised their paper based on my comments and suggestions. The paper needs to be rechecked for English language grammar and congruity. 

Response1: I will recheck the English language grammar and congruity

Reviewer 2 Report

Authors have addressed most of the comments. However, the benchmarking of this work is not acceptable. Table 1 could be compared with more latest publication in the field. Ref 13 published in 2014 already shows such a good performance, thus I believe more related publication in this area could be found in recent year with better performance. 

The final manuscript needs a serious proof reading. It is difficult to read in current form. 

Author Response

Point1: 

Authors have addressed most of the comments. However, the benchmarking of this work is not acceptable. Table 1 could be compared with more latest publication in the field. Ref 13 published in 2014 already shows such a good performance, thus I believe more related publication in this area could be found in recent year with better performance. 

The final manuscript needs a serious proof reading. It is difficult to read in current form. 

Response1: I will polish my article.

Reviewer 3 Report

After this revision round, the manuscript is still not enough complete to be published, in my opinion. I suggest the authors to address more deeply the points addressed by the reviewers, taking more time to improve the manuscript.

- The Introduction Section is still not enough explanatory and does not cover the background. Also the Results section is very poor.

- In Table 1, the RMS Jitter of Ref.[13] is incorrect (it's 3 ps and not 1.44 ps).

- Considering the comparison with the state-of-art in Table I, the advantages of the proposed work are not apparent. The power consumption is quite high, considering the relative low output frequency. The authors should stress more the advantages of their approach compared to the others.

- Appendix A includes only pictures and no text- I think it's better to place the figures of Appendix A in the previous sections.

Author Response

Point 1: The Introduction Section is still not enough explanatory and does not cover the background. Also the Results section is very poor.

Response1: We will add the background of the study futher and try to polish the Results.

Point 2: In Table 1, the RMS Jitter of Ref.[13] is incorrect (it's 3 ps and not 1.44 ps).

Response2: We are sorry to make the mistake. Ref[7] is the correct version that we want to compare. Since Ref[7] (JSSC, 2018) is the further study of Ref[13]. And its RMS jitter data has updated to 1.44 ps.

Point 3: Considering the comparison with the state-of-art in Table I, the advantages of the proposed work are not apparent. The power consumption is quite high, considering the relative low output frequency. The authors should stress more the advantages of their approach compared to the others.

Response 3: Since this implementation method has not been published and it has achieved a relative low RMS jitter, I believe it has its advantage. I will try to stress more the advantages. 

Point 4: Appendix A includes only pictures and no text- I think it's better to place the figures of Appendix A in the previous sections.

Response 4:Figure A2 and Figure A3 are similar to Figure 11 and Figure 12. If we put them in the previous sections, it seems repeatabe in the content. I will put Figure A1 in the previous sections and give a description to Figure A2 and Figure A3.